

# Physical activity and health-related quality of life among adults living in Jeddah city Saudi Arabia

Ola Akram Abdulrashid[1], Hassan Bin Usman Shah[2],
Wijdan Abdulkareem Baeshen[3], Sarah Mohammad Aljuaid[3],
Enas Awad Alasmari[3], Rania Ali Baokbah[3], Reema Ali Baokbah[3],
Nojoud Mohammed Alamoudi[1], Maha Saleh Alkhelewi[1],
Amal Abdullah Turkistani[1], Ahmed Abdullah Alharbi[1],
Abdulrehman Ahmed Alghamdi[1], Fawaz Alharthi[1],
Mohammad Alcattan[1] and Amnah Marwan Haikal[4]

[1] Directorate of Health Affairs, Planning and Transformation Office, Jeddah, Saudi Arabia
[2] The Kirby Institute, University of New South Wales, Sydney, New South Wales, Australia
[3] Directorate of Health Affairs-Jeddah Research and Studies Department, Jeddah, Saudi Arabia
[4] Directorate of Health Affairs, Academic Affairs, Jeddah, Saudi Arabia

Corresponding author
Hassan Bin Usman Shah,
hassanbinusman@hotmail.com

## ABSTRACT

**Background:** Physical activity can improve health-related quality of life (HRQoL) in adults. However, the effect of physical activity on quality of life is unclear among the Saudi adult population. The study aimed to determine the association between physical activity and HRQoL in apparently healthy adults in Jeddah, Saudi Arabia.

**Methods:** This cross-sectional study was conducted among visitors of shopping malls, walking tracks/fitness centres/gyms and governmental hospitals in Jeddah from March to September 2022. Physical activity was measured with the electronic template of the general practice physical activity questionnaire (GPPAQ), while the HRQoL was measured using the 36-Item Short Form Survey (SF-36) questionnaire. A multiple linear regression model investigated the relationship between physical activity and HRQoL.

**Results:** A total of 693 individuals participated in this study, with a mean age of 36 (±11). Individuals who fall in the active category of the physical activity level were mostly younger men (37% vs 21%), were non-smokers (30% vs 10%) and had no comorbid condition (29% vs 15%). After adjusting for covariates, men (adjusted β 4.43, 95% CI [1.44–7.41]) with higher physical activity levels (active-adjusted β 10.11, 95% CI [5.44–14.77]) had better scores on the physical component summary (PCS). Similarly, mental component summary (MCS) scores for men (adjusted β 6.51, 95% CI [3.40–9.63]) and physical activity levels (active-adjusted β 9.77, 95% CI [4.90–14.64]) were high.

**Conclusion:** The article reinforces how physical activity contributes towards each dimension of HRQoL. Physical inactivity is a growing public health challenge in Saudi Arabia, affecting all age groups. Investing in innovative strategies and establishing targeted health education programs for academic institutions and communities are required to enhance healthy habits. Additionally, constructing more local sports facilities and concession packages, especially for the females at the gyms, can motivate individuals and promote physical activity.

# INTRODUCTION

Physical activity (PA) benefits people in terms of physical and psychological health (*An et al., 2020*; *Warburton, Nicol & Bredin, 2006*). By definition, physical activity is any skeletal muscle body movement or activity that results in energy expenditure above the basal level (*World Health Organization, 2017*). Physical inactivity, slowly being recognised as a global pandemic, is one of the most prevalent modifiable risk factors for acquiring disease that results in 6–9% of deaths worldwide (*Ding et al., 2016*; *Hallal et al., 2012*). Rapid changes in lifestyle, increasing use of technology and motorised transport, changing dietary patterns, and increased tobacco use are major contributory factors prevalent in all societies and countries, including Saudi Arabia (*Khalaf et al., 2013*; *World Health Organization, 2018*). Despite the well-acknowledged and widely known benefits linked to PA and World Health Organisation (WHO) recommendations, fewer than a third of adults comply with these guidelines (*World Health Organization, 2017*; *World Health Organization, 2018*).

Health-related quality of life (HRQoL) is a global construct often used interchangeably with an individual's subjective self-perception of personal well-being and other aspects of their current health status (*Marker, Steele & Noser, 2018*). HRQoL is a multidimensional concept, with subdomains to assess the physical (*e.g.,* evaluation of physical functions) and mental (*e.g.,* emotional health) well-being in different populations (*Marquez et al., 2020*). HRQoL can provide a holistic understanding of the subjective disease burden associated with individual health and help improve overall disease management (*Lubans et al., 2016*; *Marker, Steele & Noser, 2018*). Being physically active has been suggested to enhance HRQoL and directly impact well-being (*Kantor, Grimes & Limbers, 2015*; *Marker, Steele & Noser, 2018*).

The Kingdom of Saudi Arabia also mirrors the alarming trends of physical inactivity (*Al-Hazzaa, 2004*; *Alasqah et al., 2021*). Indeed, recent data show physical activity prevalence ranged between 4% to 45%, depending on gender, age, location, and target population in Saudis (*Alasqah et al., 2021*). Unfortunately, the reported inactivity is prevalent mainly among children and young adults (*Alqahtani et al., 2021*), with a 30% decrease in physical activity levels from childhood to young adults (*Al-Hazzaa, 2004*). Significant transformations in lifestyle, excessive use of technology, motorised transport, a change in eating habits from a traditional pure diet to unhealthy junk food (*Mandoura et al., 2017*), and cultural norms are among some of the barriers behind inactivity. Limited physical activity opportunities for females and these risk factors tend to cluster, resulting in more non-communicable diseases and obesity among females (*Almalki, Ibraheem & Alotibi, 2021*; *Khalaf et al., 2013*). The high prevalence of inactivity in the Saudi adult population represents a major public health concern with escalating disease burden (*Alasqah et al., 2021*). This study aimed to determine how physical activity improves each HRQoL domain and identify the factors associated with HRQoL among adults living in Jeddah, Saudi Arabia. Highlighting gaps and barriers to physical activity and its effect on

HRQoL at the population level can potentially reduce the burden of physical inactivity-related diseases, thereby informing health service delivery and policy.

## METHODS

### Study setting, study population and sampling technique

This cross-sectional interview-based study was conducted in three shopping malls, three walking tracks and fitness centres/gyms and three governmental hospitals in Jeddah, Kingdom of Saudi Arabia. Multistage stratified random sampling technique was used to identify these shopping malls, walking tracks and fitness centres/gyms and hospitals. The study duration was 6 months, from March to September 2022.

Jeddah is divided into five administrative units, each having a tertiary care hospital, malls and many gyms. Three administrative units were initially selected using a lottery method. These three areas exhibit significant socioeconomic differences in their population. From each selected administrative unit, one hospital, one gym or a walking track and one shopping mall were selected using a simple random selection approach. Individuals coming to each selected shopping mall and visitors of admitted patients coming to the hospital who were willing to participate were selected through systematic sampling with a random start approach (every 3rd person). Similarly, regular gymgoers (attending the gym at least twice a week for the last 2 months) were randomly selected. A minimum duration of 2 months was considered since many gyms offer a month of the free trial, and people usually do not follow up.

The sample size was calculated using the online Epitools sample size calculator. Based on a study done in Al Khobar (Al-Shehri et al., 2008), Saudi Arabia, the mean physical component score was 41.3, with a standard deviation of 6. Using these figures, keeping the significance level at 95%, the desired precision at 0.5; the calculated sample size was 546. We increased the sample by 25% to 685 for any dropouts/incomplete forms.

### Study tool, data collection & scoring

After collecting the demographic profile data, a widely used, reliable and validated instrument, 36-Item Short Form Survey (SF-36), was used to gather the data in this survey. A questionnaire translated into Arabic was adapted from a study conducted in the Jazan region (Sheikh et al., 2015) for this study. The general practice physical activity questionnaire (GPPAQ) was used to assess the physical activity level.

Before the start of data collection, 1-day training on the use of SF-36 and GPPAQ was carried out. A four-member team was trained for the data collection, including a doctor, a female nurse and two volunteers (a male and a female) from the Research and Studies department, Ministry of Health (MoH). Keeping in view the cultural norms, only female team members collected data from gyms at the ladies-only time and from visitors in the female wards.

Visitors were approached and explained the purpose of this campaign. Details on the demographics and the SF-36 questionnaire items were collected from the participants. The participant's physical activity level was assessed using the electronic version of the

GPPAQ with a tablet provided by MoH (available at https://www.gov.uk/government/publications/general-practice-physical-activity-questionnaire-gppaq), and the physical activity index was calculated electronically. The physical activity index categorises the participants into active, moderately active, moderately inactive, and inactive (Table S1). Each interview took around 25–30 min, which was communicated and agreed upon in advance. At the end of the interview, a 5–10 min health education session was given regarding the importance of physical activity and a healthy diet using Information, Education and Communication (IEC) material approved and provided by MoH.

The SF-36 is a brief questionnaire that generates scores across eight dimensions of health. The items include; physical functioning (10 questions); physical role limitations (four questions); bodily pain (two questions); general health perceptions (five questions); energy/vitality (four questions); social functioning (two questions); emotional role limitations (three questions) and mental health (five questions). Patients were asked to rate their responses using 2-, 3-, 5-, or 6-point polytomous response options. For each dimension, a scoring algorithm was used to convert the raw scores item to a scale ranging from zero (worst health) to 100 (best health). Additionally, the SF-36 domain scores were summed together to calculate the two-component summary score; physical component summary (PCS) and mental component summary (MCS).

Data on the current smoking status (cigarette and/or sheesha) was taken. Current smokers were defined as having smoked twice or more in the last week (*Abdulrashid et al., 2018*). Participants were identified to have a comorbid condition if they had a history of diabetes, hypertension, cardiovascular diseases, renal diseases, joint/back pains or any diagnosed autoimmune disease or if they were currently on any medication. Considering it a sensitive issue and because of cultural values, the history of any other form of addiction or drug dependence was not asked.

## Outcome variables

The primary outcome of interest was the health-related quality of life (HRQoL) and the effect of physical activity on the HRQoL. HRQoL was inferred by calculating means for the eight domains and two-component summary score of SF-36. The secondary outcome was to determine factors associated with the PCS and MCS global scores of HRQoL.

## Selection criteria

- Inclusion criteria: Shopping mall and hospital visitors and gym/walking track users between 18–70 years old and willing to participate.
- Exclusion criteria: Visitors on wheelchairs or clutches, pregnant females, who had undergone any surgery in the recent past (within the last 2 months), or people with special needs were excluded.
- Individuals identified as having any mental health issues or drug dependency.

## Data analysis

Data analysis was done using Stata 16.1 (Stata Corp, College Station, TX, USA). First, we calculated descriptive statistics, including means and standard deviations for continuous variables and frequencies with percentages for categorical variables. We described demographic characteristics among all people by the level of their physical activity. Mean scores for all the SF-36 eight domains and PCS and MCS scores by physical activity levels were calculated according to the scoring guidelines (https://www.rand.org/health-care/surveys_tools/mos/36-item-short-form/scoring.html). Additionally, PCS and MCS mean scores by physical activity levels were calculated. The higher score indicates greater HRQoL (*Patel, Donegan & Albert, 2007*). To reduce the potential for gyms/ walking tracks individuals to skew the physical activity results, a stratified analysis calculating the mean PCS scores (of all three subgroups) was carried out using ANOVA with a Bonferroni correction test.

Unadjusted and adjusted linear regression analyses with 95% CI were performed to evaluate factors associated with PCS and MCS global scores. In the unadjusted analysis, variables that did not violate the assumption of proportionality were considered for inclusion in the adjusted analysis. Covariates included age groups (18–29, 30–44, 45–60 and 60+), sex, nationality, educational and marital status, history of smoking and comorbid conditions, interview location, and physical activity levels.

## Ethics statement

Ethical approval was granted by the Ministry of Health and Directorate of Health Affairs Jeddah (H-02-J-002-A01448) and the administration of shopping malls, hospitals and gym/ fitness centres. Before the interview, verbal consent was taken from the visitors and confidentiality of data was ensured.

# RESULTS

## Study participants

A total of 693 adults participated in this study, with a mean age of 36 (±11) years. Overall, 59% were females, 88% were Saudi nationals, 62% had education up to a bachelor's or a diploma, and around 58% were married. Approximately 9% were current smokers, and 6% had a comorbid condition. The majority of the study participants were interviewed in shopping malls (48%), followed by gyms/walking tracks (31%) and hospital visitors (21%) (Table 1).

The number of active or moderately active people was around 28% and 34%, respectively. Compared to the inactive or moderately inactive people, individuals who were active or moderately active were mostly men (70% *vs* 30%) and were single (68% *vs* 32%). A higher proportion of active or moderately active people had a bachelor's or a higher degree (60% *vs* 40%) and were frequent gym/walking track users (77% *vs* 23%). The majority of the individuals with any comorbid condition and who had a history of smoking were also inactive (Table 1).

**Table 1 Demographic characteristics of participants in Jeddah, by physical activity levels, 2022, *n* = 693.**

| Characteristics, *n* % | Total *n* (%) | Active *n* (%) | Moderately active *n* (%) | Moderately inactive *n* (%) | Inactive *n* (%) |
|---|---|---|---|---|---|
| **Total** | ***n* = 693** | ***n* = 193 (28%)** | ***n* = 238 (34%)** | ***n* = 157 (23%)** | ***n* = 105 (15%)** |
| Sex | | | | | |
| Male | 287 (41) | 106 (37) | 95 (33) | 50 (17) | 36 (13) |
| Female | 406 (59) | 87 (21) | 143 (35) | 107 (26) | 69 (17) |
| Age group | | | | | |
| 18–29 | 206 (30) | 63 (31) | 68 (33) | 48 (23) | 27 (13) |
| 30–44 | 355 (51) | 102 (29) | 134 (38) | 75 (21) | 44 (12) |
| 45–60 | 110 (16) | 26 (24) | 33 (30) | 29 (26) | 22 (20) |
| >60 | 22 (3) | 2 (9) | 3 (14) | 5 (23) | 12 (55) |
| Nationality | | | | | |
| Saudi | 609 (88) | 170 (28) | 208 (34) | 141 (23) | 90 (15) |
| Non-Saudi | 84 (12) | 23 (27) | 30 (36) | 16 (19) | 15 (18) |
| Educational status | | | | | |
| Up to high school | 169 (24) | 54 (32) | 53 (31) | 33 (20) | 29 (17) |
| Bachelor's/Diploma | 427 (62) | 112 (26) | 144 (34) | 105 (25) | 66 (15) |
| Masters and above | 97 (14) | 27 (28) | 41 (42) | 19 (20) | 10 (10) |
| Marital status | | | | | |
| Single | 237 (34) | 63 (26) | 99 (42) | 52 (22) | 23 (10) |
| Married | 404 (58) | 119 (29) | 126 (31) | 92 (23) | 67 (17) |
| Divorced | 42 (6) | 9 (21) | 11 (26) | 10 (24) | 12 (29) |
| Widow | 10 (2) | 2 (20) | 2 (20) | 3 (30) | 3 (30) |
| Location | | | | | |
| Gym/Walking tracks | 214 (31) | 87 (41) | 76 (36) | 35 (16) | 16 (7) |
| Malls | 330 (48) | 75 (23) | 114 (35) | 81 (25) | 60 (18) |
| Hospital visitors | 149 (21) | 31 (21) | 48 (32) | 41 (28) | 29 (19) |
| Smoking (cigarette or sheesha) | | | | | |
| Yes | 63 (9) | 6 (10) | 9 (14) | 24 (38) | 24 (38) |
| No | 630 (91) | 187 (30) | 229 (36) | 133 (21) | 81 (13) |
| Comorbid condition | | | | | |
| Yes | 41 (6) | 6 (15) | 8 (20) | 15 (37) | 12 (29) |
| No | 652 (94) | 187 (29) | 230 (35) | 142 (22) | 93 (14) |

## SF-36 domain scores and the component summary scores

SF-36 scores for the eight domains were calculated, showing a higher mean for active and moderately active individuals. Additionally, the PCS and MCS scores were calculated. Compared to inactive, the mean PCS and MCS scores for the active category were 79 *vs* 65 and 71 *vs* 57, respectively (Table 2). Additional stratified analysis shows no statistical difference ($p = 0.108$) in the mean PCS scores across the three subgroups (gym/walking tracks, hospital visitors and mall visitors; 75, 71 and 72, respectively).

**Table 2  Mean SF-36 domain scores, by physical activity levels.**

| SF-36 domains | Active | Moderately active | Moderately inactive | Inactive |
|---|---|---|---|---|
| | Mean (SD) | Mean (SD) | Mean (SD) | Mean (SD) |
| Physical functioning | 81 (26) | 75 (29) | 74 (25) | 67 (30) |
| Role physical | 82 (34) | 76 (35) | 69 (37) | 60 (42) |
| Role emotional | 73 (40) | 67 (42) | 54 (44) | 55 (45) |
| Energy fatigue | 63 (19) | 60 (18) | 47 (17) | 50 (19) |
| Emotional well being | 64 (13) | 63 (14) | 56 (12) | 57 (18) |
| Social functioning | 79 (21) | 75 (21) | 65 (22) | 66 (29) |
| Bodily pain | 84 (20) | 75 (23) | 67 (21) | 69 (27) |
| General Health | 71 (15) | 69 (14) | 61 (14) | 62 (19) |
| Physical component summary (PCS) | 79 (18) | 74 (18) | 68 (17) | 65 (22) |
| Mental component summary (MCS) | 70 (18) | 66 (18) | 56 (19) | 57 (23) |

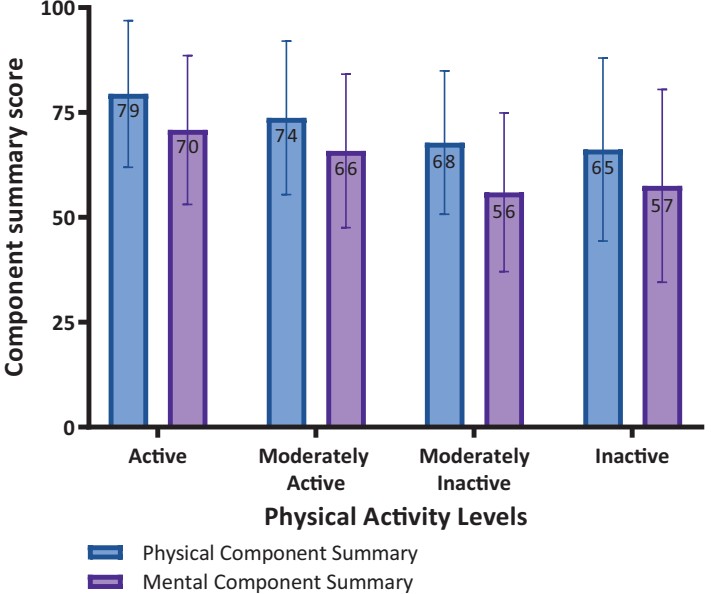

**Figure 1  Physical and mental component summary scores according to the physical activity levels.**

Component summary scores (physical and mental components) according to the physical activity levels are shown in Fig. 1. The physical component domain scores were higher in all activity levels (active 79 *vs* 70; moderately active 74 *vs* 66; moderately inactive 68 *vs* 56; and inactive 66 *vs* 58) than the mental component domain scores (Fig. 1), highlighting the importance of physical activity.

## Factors associated with physical component summary (PCS) scores and health-related quality of life

In adjusted analysis, PCS scores were significantly associated with male gender (adjusted β 4.43, 95% CI [1.44–7.41]) and physical activity levels (moderately active- adjusted β 5.73,

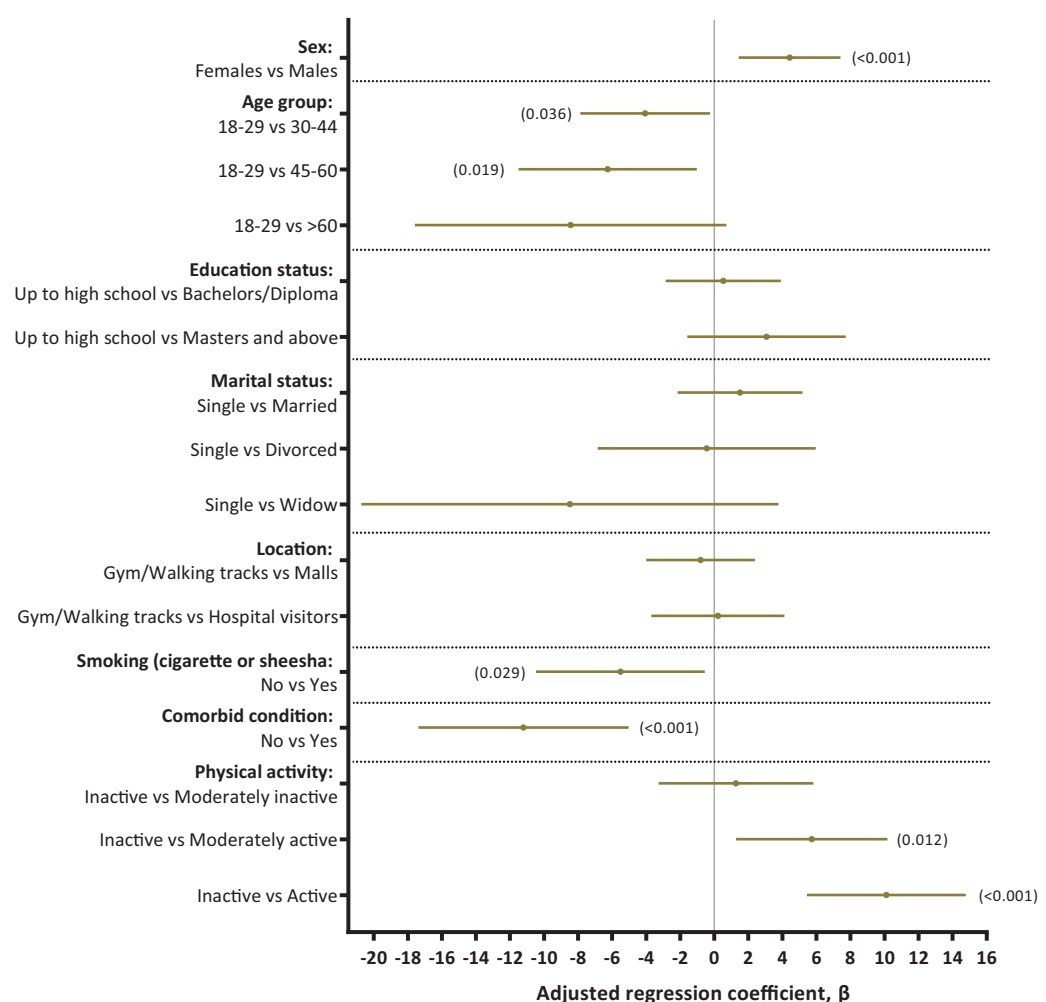

**Figure 2 Factors associated with PCS and health-related quality of life among adults in Jeddah.**

95% CI [1.28–10.17] and active- adjusted β 10.11, 95% CI [5.44–14.77]). Compared to the individuals between 18–29 years of age, individuals in age groups 30–44, 45–60 and 60+ had lower PCS scores (adjusted β −4.06, 95% CI [−7.87 to −0.25]; −6.27, 95% CI [−11.50 to −1.03]; and −8.44, 95% CI [−17.59 to 0.71]), respectively. History of smoking and having any comorbid conditions significantly affected PCS scores (adjusted β −5.52, 95% CI [−10.47 to −0.56] and −11.21, 95% CI [−17.39 to −5.03]), respectively. The adjusted R-squared value for the final model was 0.11. Marital status and nationality had no significant association with PCS (Fig. 2 and Table S2).

## Factors associated with mental component summary (MCS) scores and health-related quality of life

In adjusted analysis, MCS scores were significantly associated with male gender (adjusted β 6.51, 95% CI [3.40–9.63]) and physical activity levels (moderately active-adjusted β 7.57, 95% CI [2.92–12.21] and active-adjusted β 9.77, 95% CI [4.90–14.64]). Compared to the individuals between 18–29 years of age, individuals in age groups 30–44 had less MCS

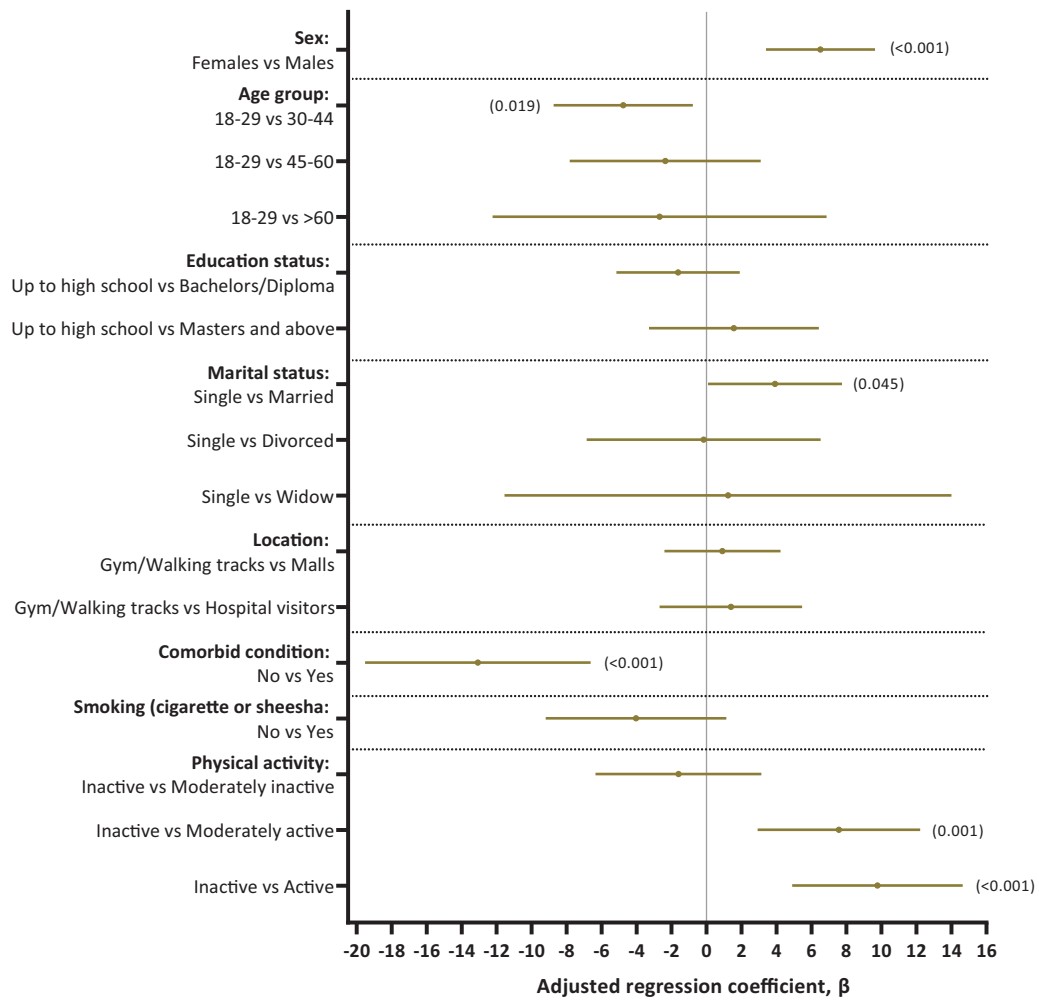

**Figure 3 Factors associated with MCS and health-related quality of life among adults in Jeddah.**

scores (adjusted β −4.76 95% CI [−8.74 to −0.78]). History of having any comorbid conditions significantly affected MCS scores (adjusted β −13.08, 95% CI [−19.52 to −6.63]). Smoking status was not significantly associated with MCS scores (adjusted β −4.03, 95% CI [−9.20 to 1.13]). The adjusted R-squared value for the final model was 0.12. Marital status and nationality had no significant association with MCS (Fig. 3 and Table S3).

# DISCUSSION

Our study demonstrated the importance of physical activity on health-related quality of life among apparently healthy adults in Jeddah, Saudi Arabia. Physically active people had higher HRQoL domain scores, enhancing their physical and mental health. Young and middle-aged men, those with a higher education level and who had no history of smoking or comorbid disease, had a higher physical activity level and better HRQoL. Statistically significant differences in HRQoL and better physical and mental health are reported among those who engage in any PA (like regular gym users or individuals using walking

tracks frequently) compared to inactive individuals. Considering the link between HRQoL and PA, educating the general public regarding the importance of physical activity and its impact on HRQoL and implementing practical initiatives for health promotion becomes a priority.

Maintaining or improving HRQoL and well-being is a universal goal; unfortunately, efforts to overcome this global physical inactivity epidemic are insufficient (*Marquez et al., 2020*). Being physically active has been suggested to enhance HRQoL and subjective well-being (*Skevington & Böhnke, 2018*). Our study reported around 62% of participants as active or moderately active, which was better than the 45% activity prevalence among Saudi adults (*Alasqah et al., 2021*). A high prevalence of lifestyle-related diseases and rising obesity levels in the Saudi population is mainly due to a large population being inactive (*Alasqah et al., 2021*). Diseases related to physical inactivity and low level of HRQoL will continue mounting, burdening heavily on the health sector cost (*Alqahtani et al., 2021*). Hence, efforts to promote physical activity become vital in the community for risk reduction of non-communicable diseases (*Torres et al., 2018*).

Similar trends in the physical activity of men and women were observed in agreement with older studies (*Gil-Lacruz et al., 2021*; *Guthold et al., 2018*; *Saffer et al., 2013*), with Saudi females being more inactive than men. Since men engage in physical activity more, the HRQoL index was better for each PCS and MCS subscale. These results are reinforced according to previous studies on this topic (*An et al., 2020*; *Liao et al., 2021*; *Marker, Steele & Noser, 2018*; *Sun, Norman & While, 2013*). A sedentary lifestyle, cultural values, scarce gyms or sporting options and limited time slots for females in our society are among a few reasons for less female involvement (*Almalki, Ibraheem & Alotibi, 2021*; *Khalaf et al., 2013*). Targeted efforts are needed to address these inequalities that ensure more female participation.

An increase in physical activity and well-being was observed in the young population. A statistically significant difference in HRQoL was observed between age groups 45 and under compared to the older population; other studies mentioned similar results (*Liao et al., 2021*; *Marquez et al., 2020*; *Skevington & Böhnke, 2018*). Although the aging population might not have the same stress from life events and economic problems compared to the young, engagement in physical activity is less (*Bláfoss et al., 2019*; *Liao et al., 2021*). A possible explanation for this is that as people grow older, their energy levels tend to decrease, and they may spend less time engaging in physical activities (*Liao et al., 2021*). Providing and creating age-friendly physical activity opportunities, especially for the elderly, should be a priority for the local government.

People with higher education levels had more health knowledge, which is an important force driving changes to and promoting physical activity (*An et al., 2020*; *Castillo et al., 2020*). In accordance with other studies (*Castillo et al., 2020*; *Gil-Lacruz et al., 2021*; *Guthold et al., 2018*), the results of our research also supported the benefits of higher education levels and exhibited the effects on physical and mental health. Surprisingly, a few studies did not support this point and showed no significant relationship between education and subjective well-being (*Portela, Neira & Salinas-Jiménez, 2013*; *Rodríguez-Pose & Von Berlepsch, 2014*). Institute-based health education sessions on the importance

of physical activity can improve health knowledge and the ability to deal with life problems (*An et al., 2020*).

Tobacco-related diseases are a leading cause of preventable deaths worldwide; unfortunately, tobacco use in its different forms is increasing among Saudi women and young people (*Abdulrashid et al., 2018*). Participants with a history of smoking were significantly associated with lower PCS scores. People who smoke are usually less active with low energy levels, and smoking itself can result in many health conditions (*Abdulrashid et al., 2018*). Understanding the factors behind the upsurge and health education can be vital in vulnerable groups (*Abdulrashid et al., 2018*; *Badran & Laher, 2020*).

In general, people who exercise regularly reported better levels of health for each domain. Spending time in physical activity is a value in itself since it contributes to both the physical and psychological development of an individual (*An et al., 2020*; *Kantor, Grimes & Limbers, 2015*). Physical health improves cardiovascular status, strengthens the body, and enhances functional capacity. Additionally, psychological improvement is through changes in anxiety and depression levels and increased self-efficacy, view of oneself, and mental health (*Siefken, Junge & Laemmle, 2019*; *Steptoe, 2019*). Innovation strategies and opportunities for people of all age groups to engage in PA activities should be a priority area.

## LIMITATIONS

There are several limitations to our study. First, some covariates were not controlled, including time spent on screen, BMI levels, socioeconomic status, time spent in the gym or walking track, and questions regarding diet/eating habits. Addiction/drug dependence are important contributors to and risk factors for poor physical and mental health. However, given the cultural norms/values, data on these factors was challenging to obtain. Second, recall bias for chronic diseases and medication history was possible. Third, causal relationships cannot be made, given the study's cross-sectional design. Although we had a few limitations in our study design, the strength of this study was its fairly diverse group of community members involving all socioeconomic classes, including active people from the gym/walking tracks. We suggest an in-depth interview with the females to highlight the factors behind less physical activity; it will help us understand and give us a chance to address the lack of opportunities.

## CONCLUSION

Regular physical activity impacts all the HRQoL domains, improving physical and mental health. Physical inactivity is growing in Saudi Arabia, affecting mainly females and older individuals. Considering the hypothesis that higher physical activity would be associated with better HRQoL, a focus on constructing more local sports facilities and ensuring concession packages, especially for the females at the gyms, would have benefits. Practical implications include supporting health education (school and community-based), banning or limiting access to tobacco products and establishing programs that motivate physical activity (introducing sports carnivals) for all age groups to improve HRQoL.

## ACKNOWLEDGEMENTS

The authors would like to acknowledge the Directorate of Health Affairs Jeddah, the Research and Studies department, and the Planning and Transformation Office, Ministry of Health, Jeddah, for their continuous support throughout this project.

### Funding

The authors received no funding for this work.

### Competing Interests

The authors declare that they have no competing interests.

### Author Contributions

- Ola Akram Abdulrashid conceived and designed the experiments, prepared figures and/or tables, authored or reviewed drafts of the article, and approved the final draft.
- Hassan Bin Usman Shah conceived and designed the experiments, analyzed the data, prepared figures and/or tables, authored or reviewed drafts of the article, and approved the final draft.
- Wijdan Abdulkareem Baeshen conceived and designed the experiments, authored or reviewed drafts of the article, and approved the final draft.
- Sarah Mohammad Aljuaid conceived and designed the experiments, authored or reviewed drafts of the article, and approved the final draft.
- Enas Awad Alasmari conceived and designed the experiments, authored or reviewed drafts of the article, and approved the final draft.
- Rania Ali Baokbah performed the experiments, authored or reviewed drafts of the article, and approved the final draft.
- Reema Ali Baokbah performed the experiments, authored or reviewed drafts of the article, and approved the final draft.
- Nojoud Mohammed Alamoudi performed the experiments, analyzed the data, authored or reviewed drafts of the article, and approved the final draft.
- Maha Saleh Alkhelewi performed the experiments, analyzed the data, prepared figures and/or tables, authored or reviewed drafts of the article, and approved the final draft.
- Amal Abdullah Turkistani performed the experiments, analyzed the data, prepared figures and/or tables, authored or reviewed drafts of the article, and approved the final draft.
- Ahmed Abdullah Alharbi performed the experiments, prepared figures and/or tables, authored or reviewed drafts of the article, and approved the final draft.
- Abdulrehman Ahmed Alghamdi performed the experiments, prepared figures and/or tables, authored or reviewed drafts of the article, and approved the final draft.
- Fawaz Alharthi performed the experiments, prepared figures and/or tables, authored or reviewed drafts of the article, and approved the final draft.

- Mohammad Alcattan conceived and designed the experiments, prepared figures and/or tables, authored or reviewed drafts of the article, and approved the final draft.
- Amnah Marwan Haikal performed the experiments, prepared figures and/or tables, authored or reviewed drafts of the article, and approved the final draft.

### Human Ethics

The following information was supplied relating to ethical approvals (*i.e.*, approving body and any reference numbers):

The Ministry of Health and Directorate of Health Affairs Jeddah.

### Data Availability

The Stata file has been added as a Supplemental File

### Supplemental Information

Supplemental information for this article can be found online at http://dx.doi.org/10.7717/peerj.16059#supplemental-information.

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
