# Peer review of "Physical activity and health-related quality of life among adults living in Jeddah city Saudi Arabia"

_PeerJ, doi:10.7717/peerj.16059_

## Round 0.1 · original submission · Major Revisions

Please fully address Reviewer 1 and Reviewer 3 comments. You may provide a response to Reviewer 2 comment but this is optional.

Reviewer 1 ·

Basic reporting

INTRODUCTION
 Line 67: Delete “;” after “factors”
 Authors defined clear and comprehensive reasoning for the completion of this study
o This study attempts to fill a gap that exists in the area of physical activity and health research within Saudi Arabia
o Acknowledges the impact advancements in technology and shifts in culture have on physical activity

Experimental design

METHODS
 Multistage stratified random sampling technique allows for low between-group variance and high within-group variance
o Beneficial for generalizability and validity
o Slight concern that sampling from a population of gym/fitness center users may slightly skew results as these individual obviously consider their physical health/fitness levels to be a priority and will on average have higher PA scores
 Sample size was determined by the online Epitools sample size calculator (n = 546) and was increased by 25% (final n = 685) to account for loss to follow up and/or incompletes
 I understand that a discussion of addiction/drug dependence goes against cultural norms/values, but it is an important contributor to and risk factor for poor physical and mental health and its exclusion calls into question the comprehensive nature of the results
 Explicit inclusion and exclusion criteria
 Multivariate regression analysis was completed and should be more explicitly mentioned in this section

Validity of the findings

RESULTS
 Line 245: Replace “less PCS” with “lower PCS”

DISCUSSION
 Limitations are adequately mentioned, but I am concerned that results from his study may not be as valid and reliable given that multiple covariates including screen time, SES, and eating habits were not controlled for/considered
 Causal relationships cannot be made given the study’s cross-sectional design
o Line 339 – 340: Rephrase “establishing conclusions regarding the cause-effect association was difficult because our methodological design is cross-sectional.”

Reviewer 2 ·

Basic reporting

The article reinforces how physical activity contributes towards each dimension of HRQoL. But the study design was too simple, not innovative enough, and didn't seem to give us much new knowledge.

Experimental design

The study design only looked at the relationship between physical activity and happiness, and could not answer the question how physical activity improves each HRQoL domain.

Validity of the findings

no comment

·

Basic reporting

This manuscript emphasizes on the importance of physical activity on public health at a geographical location that was previously unexplored. The study highlights key points that can be worked on to improve the quality of life and reduces the chances of non-communicable disease in the population. In this context, historical information that set the stage to start looking into these correlations should be cited properly to appreciate credibility of those findings several years ago (Warburton, D. E., Nicol, C. W., & Bredin, S. S. (2006). Health benefits of physical activity: the evidence. Cmaj, 174(6), 801-809). Figures should include statistical significance, P-values or NS if it is non-significant.

Experimental design

Figure 1. shows the scores across groups but it needs to be explained why it is important in the results section. Authors need to interpret the data in this figure and describe the way it supports their hypothesis along with statistical significance.

Validity of the findings

Multivariate regression analyses should be represented graphically in a more understandable fashion e.g., a dot plot or scatter plot with statistical significance details of the experiment. Tables with all the numerical values can be moved to supplemental information.

Remove line number 178 "Age more than 70 years." This is already mentioned in the first point.

---

## Round 0.2 · accepted · Accept

The authors have addressed the reviewers' concerns satisfactorily.

Reviewer 1 ·

Basic reporting

All reviewers' concerns appear to have been adequately addressed.

Experimental design

N/A

Validity of the findings

N/A

·

Basic reporting

The manuscript quality has significantly improved and almost all the comments were addressed to the satisfactory level.

Experimental design

Line 183 - should be clarified as to which criteria the information belongs 'inclusion' or 'exclusion'.

Validity of the findings

No comments.